# Improving the Reversible LSB Matching Scheme Based on the Likelihood Re-Encoding Strategy

**DOI:** 10.3390/e23050577

**Published:** 2021-05-08

**Authors:** Tzu-Chuen Lu, Ping-Chung Yang, Biswapati Jana

**Affiliations:** 1Department of Information Management, Chaoyang University of Technology, Taichung 41349, Taiwan; s10814617@cyut.edu.tw; 2Department of Computer Science, Vidyasagar University, Midnapore 721102, India; biswapatijana@gmail.com

**Keywords:** dual imaging technique, reversible data hiding, least-significant-bit matching, re-encoding technique

## Abstract

In 2018, Tseng et al. proposed a dual-image reversible embedding method based on the modified Least Significant Bit matching (LSB matching) method. This method improved on the dual-image LSB matching method proposed by Lu et al. In Lu et al.’s scheme, there are seven situations that cannot be restored and need to be modified. Furthermore, the scheme uses two pixels to conceal four secret bits. The maximum modification of each pixel, in Lu et al.’s scheme, is two. To decrease the modification, Tseng et al. use one pixel to embed two secret bits and allow the maximum modification to decrease from two to one such that the image quality can be improved. This study enhances Tseng et al.’s method by re-encoding the modified rule table based on the probability of each hiding combination. The scheme analyzes the frequency occurrence of each combination and sets the lowest modified codes to the highest frequency case to significantly reduce the amount of modification. Experimental results show that better image quality is obtained using our method under the same amount of hiding payload.

## 1. Introduction

With the rapid development of digital technology nowadays, a large amount of information is transmitted on the Internet. Although it is convenient, it is also, however, insecure. The transmitted data may be stolen by unscrupulous third parties. Consequently, to keep the seed details from being tampered with or viewed, researchers use different information-hiding techniques to conceal and cover the information transmitted over unsecured channels. There are three major requirements for information-hiding methods, as follows [1]:(1)Security: This requirement is necessary to protect the information from being detected or cracked by illegitimate third parties.(2)Imperceptibility: To improve protection, the image cannot be blurred, deformed, or distorted, and cannot be easily detected by the human eye.(3)High Capacity: The more data embedding, the higher the distortion that occurs. Making good use of every hidden space, and increasing the capacity are also important.

Information-hiding technologies can be classified into “Reversible Data Hiding (RDH)” and “Non-Reversible Data Hiding (NRDH).” Most of the early information-hiding technologies are NRDH, meaning that after extracting the confidential information, there is no way to restore it to the original image, such as with the Least Significant Bit Replacement (LSB), LSB matching, modulus methods, and so on. However, scholars have successively proposed various RDH techniques to fix this problem. Nowadays, researchers are working hard on increasing the hiding capacity and improving the visual quality, which is also the standard for measuring the quality of the suggested method [2,3].

The RDH schemes focus on the image quality and whether it can be restored in the future, so the RDH method usually uses a lower amount of secret information. Increasing the capacity of RDH storage has become one of the challenges that have been paid much attention to in recent years. Common RDH methods include the Difference Expansion, Histogram Shifting, Compression Image and Dual-Images (DI) techniques, and so on [4,5,6,7,8,9,10,11,12,13].

Some RDH methods are essentially reversible, while some become reversible after being improved by certain techniques [14,15,16,17]. Among them, the LSB matching method, which was proposed by Mielikainen in 2006, uses the parity of the Least Significant Bit value of pixels to hide the confidential information [18,19,20,21,22,23]. Although this method is a non-reversible method, Lu et al. improved it to become reversible in 2015 [24]. They found that while using LSB matching, there are seven cases of modifications that cannot be recovered successfully, resulting in the method being non-reversible. To fix this problem, Lu et al. used a mapping table, which instructs the corresponding modifying rules to make the LSB matching method become a reversible technique when they encounter the non-reversible seven cases. After this method was proposed, many investigators began to study how improvements could increase the efficiency of this method. Tseng et al.’s scheme is one of these methods, which proposes a single-pixel hiding method that effectively improves the quality of stego-images [25]. Lu et al.’s method uses two pixels as a pair for dual-image hiding. That is, four pixels are used to hide four secret bits in each hiding process. Tseng et al.’s method, on the other hand, uses only one pixel of both images at a time. In other words, only two pixels of the two images are used at a time to embed two secret bits. Hence, the number of pixel modifications is reduced.

This investigation found that the hiding performance of Tseng et al.’s scheme is determined by the frequency of the modification. If one can control the modification times, then the image quality is controllable. Hence, this study analyzes the hidden rules of Tseng et al.’s scheme. The proposed scheme calculates in advance the number of occurrences of all hidden rules and the amount of modification to re-encode the confidential information for reducing the image distortion. Consequently, the proposed method achieves the result of improving stego-image visual quality.

## 2. Literature Discussion

In 2006, Mielikainen proposed an LSB matching method to embed two secret bits into two pixels. In their scheme, only one pixel will be modified in the embedding process such that effectively reduce the image distortion. However, the scheme is non-reversible. In 2015, Lu et al. extended the LSB matching method to become a reversible hiding scheme using dual-image technique. In Lu et al.’s scheme, four secret bits are concealed into two pixel-pairs in two copy images, respectively. After that, Wang et al. enhanced Lu et al.’s scheme to reuse the second pixel of the pixel pair for concealing one more secret bit. Different from Lu et al. and Wang et al.’s scheme, Tseng et al. use one pixel instead of two pixels in the pixel pair to embed one secret bit. More details about the related works are shown below.

### 2.1. Least Significant Bit (LSB) Matching Method

The LSB matching method, which was proposed by Mielikainen in 2006, improved on the LSB replacement method by Chen et al. in 2004. The image quality of Chen et al.’s scheme is poor when the amount of confidential information increases. Therefore, Mielikainen proposed the LSB matching method to improve the image quality of Chen et al.’s scheme [22]. The hiding flowchart of LSB replacement is shown in Figure 1.

In their scheme, a cover image is divided into several 1 × 2 non-repetitive blocks, and A and B are represented as two pixels for each block. The confidential information to be hidden in each group is $m_{1}$ and $m_{2}$. The scheme follows the tree flow diagram, as shown in Figure 1, to find the proper modify rule. The LSB of A is compared with $m_{1}$, and then the value of F function is compared with $m_{2}$ to determine the final modify rule. The F function is shown as follows:(1)$$F\left(A,\;B\right)=LSB\left(\lfloor \frac{A}{2}\rfloor +B\right)$$

Four conditions of the final modify rule are shown, as follows:Case 1: When $LSB\left(A\right)=m_{1}$ and $F\left(A,\;B\right)=m_{2}$, the pixel pair A and B does not need modification.Case 2: When $LSB\left(A\right)=m_{1}$ and $F\left(A,\;B\right)\neq m_{2}$, the pixel A does not change, and B=B+1 or B=B−1.Case 3: When $LSB\left(A\right)\neq m_{1}$ and $F\left(A-1,\;B\right)=m_{2}$, the pixel A=A−1, and B does not change.Case 4: When $LSB\left(A\right)\neq m_{1}$ and $F\left(A-1,\;B\right)\neq m_{2}$, the pixel A=A+1, and B does not change.

After hiding each group of pixel pairs and confidential information in sequence, a camouflage image can be obtained.

When the receiver receives the camouflage image, the confidential information extraction process can be executed. The extraction formula is shown, as follows:(2)$$m_{1}=LSB\left(A^{\prime }\right)$$
(3)$$m_{2}=LSB\left(\lfloor \frac{A^{\prime }}{2}\rfloor +B^{\prime }\right)$$

Finally, the hiding and extracting processes are done.

Let us assume the pixel pairs to be A=128, B=130 and the confidential information to be $s={\left(01\right)}_{2}$. First, determine whether LSB(128) is the same as the confidential message $m_{1}=0$. One can observe that LSB(128) does equal 0. Therefore, the scheme uses the LSB(128, 130) function to determine whether *LSB* = $LSB\left(\lfloor \frac{128}{2}\rfloor +130\right)$ is the same as $m_{2}=1$. According to the hiding flowchart, the result is that $LSB\left(\lfloor \frac{128}{2}\rfloor +130\right)\neq 1$. Hence, the final modify rule is Case 2, where the pixel A does not change, and B=B+1. Finally, these results are $A^{\prime }=128$ and $B^{\prime }=131$. The above numerical example shows the complete hiding process, and subsequently, when the recipient is restoring, the confidential information can be restored through Equations (2) and (3), $m_{1}=LSB\left(128\right)=0$, $m_{2}=LSB\left(\lfloor \frac{128}{2}\rfloor +131\right)=1$.

### 2.2. Dual-Images Technique Based on the LSB Matching Method

Lu et al. proposed a dual-image LSB matching method in 2015 [24]. Lu et al. use a cover image to generate two copy images as the cover medium, instead of a single medium in LSB matching. Every two pixels in each copied image can hide two secret bits, or in other words, every two pixels can carry four secret bits in two copy images. This method can restore the original cover image after the extraction and recovery processes.

In the embedding procedure, the scheme copies the cover image into two identical images, then divides the original image into 1 × 2 non-repetitive blocks. The scheme sets the two pixels in the pixel group as A and B. Each group can contain four confidential messages $s=\left\{m_{1\;}m_{2\;}m_{3\;}m_{4\;}\right\}$. The scheme uses the LSB matching method to hide $m_{1\;},\;m_{2}$ into pixels $A^{\prime }$ and $B^{\prime }$ of the first camouflage image and $m_{3\;},\;m_{4\;}$ are hidden in pixels $A^{\prime \prime }$ and $B^{\prime \prime }$ of the second camouflage image. After completing the concealment, it is necessary to confirm whether the camouflage image can be restored to the original image through the averaging method. The formula of the averaging method is as follows:(4)$$A^{r}=\lfloor \frac{A^{\prime }+A^{\prime \prime }}{2}\rfloor ,\;B^{r}=\lfloor \frac{B^{\prime }+B^{\prime \prime }}{2}\rfloor$$
$A^{r}$ and $B^{r}$ represent the original value that was restored by the averaging method. When $A^{r}=A$ and $B^{r}=B$, it means that the pixel value of the camouflage image can be restored successfully without any modification. Otherwise, if $A^{r}\neq A$ or $B^{r}\neq B$, it means that the camouflage pixels are not able to be restored correctly. Therefore, the pixel values require a modification according to the seven modification rules designed by Lu et al. The seven modification rules are shown in Table 1 and the complete embedding flow chart is shown in Figure 2. In the table, $TA^{\prime }$ and $TB^{\prime }$ mean the temp stego-results of A and B in the first camouflage image. $TA^{\prime \prime }$ and $TB^{\prime \prime }$ mean the temp stego-results in the second camouflage image. After adjustment, the final camouflage pixels are modified to $A^{\prime },\;B^{\prime },\;A^{\prime \prime }$ and $B^{\prime \prime }$.

In the extraction procedure, the scheme uses the same formula as LSB matching methods, like Equations (2) and (3), to extract the secret information. The original image can be recovered by using the averaging method in Equation (4). Consider the original pixels *A* = 37 and *B* = 33 as an example. First, the scheme duplicates the image into two images of identical size. The pixel pairs are $\left(A_{1},B_{1}\right)=\left(A_{2},B_{2}\right)=\left(37,33\right)$ and set the secret data to $s={\left(0000\right)}_{2}$. According to the LSB matching method $LSB\left(A_{1}\right)=LSB\left(37\right)=1$, and because $LSB\left(37\right)\neq m_{1}=0$, it is replaced by 37−1=36. The scheme puts 36 and $B_{1}=33$ into *F* function to obtain F(36,33)=LSB(51)=1. The value of the *F* function is not equal to $m_{2}=0$, and the result is Case 4, which is $TA^{\prime }=A_{1}+1$ and $TB^{\prime }=B_{1}$.

The pixel pair in the second image $\left(A_{2},B_{2}\right)$ hides the secret data $s={\left(00\right)}_{2}$. After calculation, we obtain the same result, Case 4, which is $TA^{\prime \prime }=A_{2}+1$ and $TB^{\prime \prime }=B_{2}$. Second, the scheme checks whether the camouflage pixels can be recovered or not by referring to the seven modification rules. Here, we can see that Rule 7 of the seven modification rules confirms this, hence, the final camouflage pixels will be modified into $A^{\prime }=37-1=36,\;A^{\prime }=37-1=36$, $B^{\prime }=33-1=32$, $A^{\prime \prime }=37+1=38$ and $B^{\prime \prime }=33+2=35$.

The extracting procedure can be implemented by using Equations (2) and (3). The scheme calculates the message $m_{1}=LSB\left(36\right)=0$ and $m_{2}=LSB\left(\lfloor \frac{36}{2}\rfloor +32\right)=LSB\left(50\right)=0$ from the first pixel pair (36, 32). The messages $m_{3}=LSB\left(38\right)=0$ and $m_{4}=LSB\left(\lfloor \frac{38}{2}\rfloor +35\right)=LSB\left(54\right)=0$ are extracted from the second pixel pair (38, 35). The complete confidential message is $s={\left(0000\right)}_{2}$.

### 2.3. Improved Dual Image-Based RDH Using LSB Matching

In 2017, Wang et al. proposed an improved method of Lu et al.’s dual-image scheme [26]. The embedding process is roughly the same as that of Lu et al.’s scheme, but the difference is that after hiding two pixels, the scheme will then determine whether the second pixel can be used to conceal once again. Wang et al. proposed a new modified rule table to fix the recovery problem. The diagram of their scheme is shown in Figure 3. The formula to check whether the second pixel could be used once again is as follows:(5)$$\left|B^{\prime }-B^{\prime \prime }\right|==0$$

In Equation (5), $B^{\prime }$ represents the second pixel of the first camouflage pixel pair, and $B^{\prime \prime }$ represents the second pixel of the second camouflage pixel pair. The equation means that if the second pixels in both first and the second camouflage images are the same, then the second pixel can be used once again.

The scheme takes two pixels as a pair and uses LSB matching to embed secret data. The detailed process is described as follows:Duplicate the original image into two identical images.Use LSB matching to embed secret data $m_{1}m_{2}$ into the first image to generate $\left\{TA^{\prime },TB^{\prime }\right\}$ and secret data $m_{3}m_{4}$ into the second image to generate $\left\{TA^{\prime \prime },TB^{\prime \prime }\right\}$.Look up Wang et al.’s modification rule table, as shown in Table 2, and modify the pixels that cannot be recovered properly. There are 11 cases in the modified rule table.Use Equation (5) to check whether the second pixel can possibly be used to embed overlapping. If Equation (5) is satisfied, then the scheme uses the second pixel of the pixel pair as the first pixel to be hidden next time. Otherwise, if Equation (5) is not satisfied, the scheme skips the second pixel and finds a new pixel pair to embed the next occurrence of secret data.Repeat Steps (2) to (4) until all secret data are embedded into two camouflage images.

The extracting procedure can be implemented by using Equation (6) to extract the first secret data and using Equation (7) to extract the second secret data. Similarly, the scheme uses Equations (8) and (9) to extract the third and fourth secret data. In the extracting procedure, the scheme needs to use Equation (5) to check whether the next pixel pair is overlapping or not. If the value of $\left|B^{\prime }-B^{\prime \prime }\right|$ equals 0 or lower than 3, then the scheme uses the second pixel to perform the next extraction. Otherwise, the scheme uses the new pixel pair to extract secret data. Finally, Equation (10) is used to recover to the original pixel.
(6)$$m_{1}=LSB\left(A^{\prime }\right)$$
(7)$$m_{2}=LSB\left(\lfloor \frac{A^{\prime }}{2}\rfloor +B^{\prime }\right)$$
(8)$$m_{3}=LSB\left(A^{\prime \prime }\right)$$
(9)$$m_{4}=LSB\left(\lfloor \frac{A^{\prime \prime }}{2}\rfloor +B^{\prime \prime }\right)$$
(10)$$A=\lfloor \frac{A^{\prime }+A^{\prime \prime }}{2}\rfloor$$

### 2.4. Tseng’s Dual Image-Based RDH on the Modified LSB Matching Method

In Lu et al.’s method, the seven modified rules are implemented to fix the cases, which are not recoverable to recover correctly [25]. In the modified rules, the greatest distortion made by each pixel is 2, which might be larger than the one produced by the regular LSB matching method. In 2019, Tseng et al. tackled this issue by changing the LSB matching embedding process. Instead of using two neighboring pixels, Tseng et al. used one pixel to make a pixel pair for hiding data by implementing the modified LSB matching. The camouflage pixels generated by modified LSB matching can be recovered correctly using the averaging method without any adjustment. Relying on these changes, Tseng et al.’s method can improve the image quality and achieve retrieval without using the rule table for modification. The embedding procedure is shown in Figure 4.

There are also four cases of modifications in the modified LSB matching, which are like the original LSB matching, with only Case 3 being different. This change is to ensure the recovery procedure can be executed successfully. Four conditions of the final modify rule by Tseng et al. are shown, as follows:Case 1: When $LSB\left(A\right)=m_{1}$ and $F\left(A,A\right)=m_{2}$, the pixel pair $A^{\prime }$ and $A^{\prime \prime }$ does not need modification.Case 2: When $LSB\left(A\right)=m_{1}$ and $F\left(A,A\right)\neq m_{2}$, the pixel $A^{\prime }=A$, and $A^{\prime \prime }=A+1$.Case 3: When $LSB\left(A\right)\neq m_{1}$ and $F\left(A-1,A\right)=m_{2}$, the pixel $A^{\prime }=A+1$, and $A^{\prime \prime }=A-1$.Case 4: When $LSB\left(A\right)\neq m_{1}$ and $F\left(A-1,A\right)\neq m_{2}$, the pixel $A^{\prime }=A+1$, and $A^{\prime \prime }=A$.

## 3. Research Methods

Tseng et al.’s scheme is efficient. In their method, the maximum modification is ±1 according to the adjustment rules of LSB matching. The image quality of Tseng et al.’s scheme is very good.

In Tseng et al.’s scheme, each pixel depending on the secret message there might have several different cases. For an even pixel, its corresponding F function value might be 0 or 1. Each F function value has four different hiding cases 00, 01, 10, and 11. An odd pixel also has the same situation. Therefore, there are 2 (even or odd pixel) × 2 (F value 0 or 1) × 4 (secret messages 00, 01, 10, 11) =16 different hiding cases.

If we could set up the most frequent case to have the lowest modified values, then the image distortion could be further reduced, such that the image quality could be improved.

Hence, the proposal tries to minimize the number of cases when the modified pixel values differ by 2. It is done by redefining the four maps above so that the worst case is used a minimal number of times. The new maps should be transferred to the receiver additionally to the stego-images. This study analysis and these statistics utilize the appearance rules of Tseng et al.’s scheme to adjust the modification rule table. The diagram of the proposed scheme is shown in Figure 5.

### 3.1. Preprocessing Procedure

By analyzing Tseng et al.’s scheme, we can find that a different parity of the pixel will cause different embedding results and will use different matching rules. In other words, the odd or even pixel value will affect its embedding orientation. For example, if A is an even number, then F(A, A) is the same as F(A+1, A), and it has eight different possible results. On the other hand, if A is an odd number, the results of F(A, A) are opposite of F(A+1, A), and it also has eight different possible results. For example, suppose that A=12 is an even number, the value of $F\left(A,\;A\right)=F\left(12,\;12\right)=LSB\left(\lfloor \frac{12}{2}\rfloor +12\right)=0$ is equal to $F\left(A+1,\;A\right)=F\left(13,\;12\right)=LSB\left(\lfloor \frac{13}{2}\rfloor +12\right)=0$. On the contrary, suppose that A=11 is an odd number, the value of $F\left(A,\;A\right)=LSB\left(\lfloor \frac{11}{2}\rfloor +11\right)=0$ is different from $F\left(A+1,\;A\right)=F\left(12,\;11\right)=LSB\left(\lfloor \frac{12}{2}\rfloor +11\right)=1$. Therefore, we can ignore the modification of F(A+1, A) and just look at the part of F(A, A). The analysis of F(A, A) is shown in Table 3.

In the first part, if A is an even number, then there are two different cases of F(A, A) for embedding the secret message, where F(A, A)=0 or F(A, A)=1. Furthermore, for each case of F(A, A) there are four different embedding situations with $m_{1}$ and $m_{2}$. Therefore, there are eight different cases applicable to an even number. For example, suppose that A=12, then LSB(A)=LSB(12)=0 and the value of F(A, A)=F(12, 12)=LSB(⌊122⌋+12)=0. If $m_{1}=1$ and $m_{2}=1$, then $A^{\prime }=A+1=13$ and $A^{\prime \prime }=A-1=11$. The image distortion made by this case is $\delta ={\left(A^{\prime }-A\right)}^{2}+{\left(A^{\prime \prime }-A\right)}^{2}={\left(13-12\right)}^{2}+{\left(11-12\right)}^{2}=2$. There are also eight different cases for an odd number. For example, suppose that A=9, then LSB(A)=LSB(9)=1 and the value of F(A, A)=F(9, 9)=LSB(⌊92⌋+9)=1. If $m_{1}=1$ and $m_{2}=0$, then $A^{\prime }=A=9$ and $A^{\prime \prime }=A+1=10$. The image distortion made by this case is $\delta ={\left(A^{\prime }-A\right)}^{2}+{\left(A^{\prime \prime }-A\right)}^{2}={\left(9-9\right)}^{2}+{\left(9-10\right)}^{2}=1$.

For the even number with LSB(A) and F(A, A)=0, the distortions for the message bits $\left(m_{1},m_{2}\right)$ with (0,0), (0,1), (1,0), and (1,1) are 0, 1, 1, and 2, respectively. The scheme ranks the four cases according to their corresponding distortion value. The column “Ordering (ρ)” in Table 3 shows the ordering results. The maximum distortion δ=2 is made by the cases with ρ=3. In other words, if the occurrence frequency of the cases with ρ=3 is large then the image quality becomes worse.

Before the embedding procedure, the proposed scheme analyzes the occurrence frequency of every combination between the cover image and the secret message, for generating a re-encoding table. In the analysis processing, a cover pixel A is duplicated to generate two temp pixels for concealing two message bits $m_{1}$ and $m_{2}$. In Table 4, the symbol γ is the frequency of the combination, δ is the distortion made by the modification, ρ is the original order sorting by δ, and θ is the total distortion computed by θ=γ×δ. The scheme re-orders the combination according to γ in a decreasing order to get the new order $\hat{\rho }$ of each combination.

In Table 4, $\gamma_{F\left(A,\;A\right)}^{LSB\left(A\right)}\left(t\right)$ means the frequency of the combination with LSB(A) and F(A, A) in the tth case, where 0≤t≤3. For example, $\gamma_{0}^{0}\left(3\right)$ is the frequency for an even number, which is LSB(A)=0 and F(A, A)=0 in the case where t=3, which results in $m_{1}=1$ and $m_{2}=1$. $\delta_{F\left(A,\;A\right)}^{LSB\left(A\right)}\left(t\right)$ is the distortion computed by ${\left(A^{\prime }-A\right)}^{2}+{\left(A^{\prime \prime }-A\right)}^{2}$ with LSB(A) and F(A, A) in the tth case. $\rho_{F\left(A,\;A\right)}^{LSB\left(A\right)}\left(t\right)$ is the ranking of the combination that is sorted by $\delta_{F\left(A,\;A\right)}^{LSB\left(A\right)}\left(t\right)$ in decreasing order. The scheme re-ranks the combination to get the new order ${\hat{\rho }}_{F\left(A,\;A\right)}^{LSB\left(A\right)}\left(t\right)$ by sorting $\gamma_{F\left(A,\;A\right)}^{LSB\left(A\right)}\left(t\right)$ in decreasing order.

An example is shown in Table 5. In the table, the occurrence frequency of the even pixel that conceals $\left(m_{1},\;m_{2}\right)=\left(1,\;0\right)$ is $\gamma_{F\left(A,\;A\right)=0}^{LSB\left(A\right)=0}\left(2\right)=135$. The distortion made by the modification rule, which is shown in Table 3, is $\delta_{F\left(A,\;A\right)=0}^{LSB\left(A\right)=0}\left(2\right)={\left(+1\right)}^{2}+{\left(0\right)}^{2}=1$. There are 135 pixels, which have the same attributes. Hence, the total number of the distortion is $\theta_{F\left(A,\;A\right)=0}^{LSB\left(A\right)=0}\left(2\right)=\gamma_{F\left(A,\;A\right)=0}^{LSB\left(A\right)=0}\left(2\right)\times \delta_{F\left(A,\;A\right)=0}^{LSB\left(A\right)=0}\left(2\right)=135\times 1=135$. Because $\gamma_{F\left(A,\;A\right)=0}^{LSB\left(A\right)=0}\left(2\right)=135\geq \gamma_{F\left(A,\;A\right)=0}^{LSB\left(A\right)=0}\left(0\right)=50\geq \gamma_{F\left(A,\;A\right)=0}^{LSB\left(A\right)=0}\left(3\right)=45\geq \gamma_{F\left(A,\;A\right)=0}^{LSB\left(A\right)=0}\left(1\right)=40$, the order of the combination, with LSB(A)=0 and F(A, A)=0, becomes ${\hat{\rho }}_{F\left(A,\;A\right)=0}^{LSB\left(A\right)=0}\left(t\right)=\left\{1,\;3,\;0,\;2\right\}$. The case t=2 is the highest frequency case, hence, the new order of the combination is the smallest value, where ${\hat{\rho }}_{F\left(A,\;A\right)=0}^{LSB\left(A\right)=0}\left(2\right)=0$. The first case is the second frequency case, so the new order is ${\hat{\rho }}_{F\left(A,\;A\right)=0}^{LSB\left(A\right)=0}\left(0\right)=1$.

The mapping of ρ and $\hat{\rho }$ of the re-encoding table is used in the embedding and extraction processes.

### 3.2. Embedding Process

The proposed scheme uses one cover pixel *A* to embed two message bits $m_{1},m_{2}$ in each embedding procedure. To reduce the total amount of distortion, the highest frequency combination is assigned the rule, which has the least distortion. Hence, the scheme modifies the pixel by using the altered rules in Table 6. Let $L{A^{\prime }}_{F\left(A,\;A\right)}^{LSB\left(A\right)}\left(\hat{\rho }\right)$ be the first rule used to modify the first camouflage pixel $A^{\prime }$, and $L{A^{\prime \prime }}_{F\left(A,\;A\right)}^{LSB\left(A\right)}\left(\hat{\rho }\right)$ be the second rule to modify the second camouflage pixel $A^{\prime \prime }$. The symbol $\hat{\rho }$ is the new order of the combination.

The scheme uses A to compute LSB(A) and F(A, A) along with $m_{1}$ and $m_{2}$ to map the re-encoding table for obtaining the new order ${\hat{\rho }}_{F\left(A,\;A\right)}^{LSB\left(A\right)}\left(t\right)$. The alter rules are $L{A^{\prime }}_{F\left(A,\;A\right)}^{LSB\left(A\right)}\left(\varepsilon \right)$ and $L{A^{\prime \prime }}_{F\left(A,\;A\right)}^{LSB\left(A\right)}\left(\varepsilon \right)$, where $\varepsilon ={\hat{\rho }}_{F\left(A,\;A\right)}^{LSB\left(A\right)}\left(t\right)$. For example, if we assume that the pixel is even, then A=14, LSB(A)=0, and F(A, A)=1. The alter rules for ε being equal to 0, 1, 2, and 3 are $\left(L{A^{\prime }}_{F\left(A,\;A\right)=1}^{LSB\left(A\right)=0}\left(\varepsilon \right),L{A^{\prime \prime }}_{F\left(A,\;A\right)=1}^{LSB\left(A\right)=0}\left(\varepsilon \right)\;\right)=\left(+0,+0\right),\left(+0,+1\right),\;\left(+1,+0\right),\;\mathrm{and}\;\left(-1,+1\right)$, respectively.

For the other example, let us assume that an odd pixel is A=15, LSB(A)=1 and F(A, A)=0. The alter rules then are (+0, +0), (+1, +0,), (+0, +1), and (−1, +1), respectively.

Following the same example shown above, the new order of the even pixel that conceals $\left(m_{1},\;m_{2}\right)=\left(1,\;0\right)$ is ${\hat{\rho }}_{F\left(A,\;A\right)=0}^{LSB\left(A\right)=0}\left(2\right)=0$. Because $\varepsilon ={\hat{\rho }}_{F\left(A,\;A\right)=0}^{LSB\left(A\right)=0}\left(2\right)=0$ the altered rule is $\left(L{A^{\prime }}_{F\left(A,\;A\right)=0}^{LSB\left(A\right)=0}\left(0\right),L{A^{\prime \prime }}_{F\left(A,\;A\right)=0}^{LSB\left(A\right)=0}\left(0\right)\;\right)=\left(+0,+0\right)$. The new distortion is ${\hat{\delta }}_{F\left(A,\;A\right)=0}^{LSB\left(A\right)=0}\left(2\right)={\left(0\right)}^{2}+{\left(0\right)}^{2}=0$. The total number of distortions becomes ${\hat{\theta }}_{F\left(A,\;A\right)=0}^{LSB\left(A\right)=0}\left(2\right)=\gamma_{F\left(A,\;A\right)=0}^{LSB\left(A\right)=0}\left(2\right)\times {\hat{\delta }}_{F\left(A,\;A\right)=0}^{LSB\left(A\right)=0}\left(2\right)=135\times 0=0$. The image distortion is reduced from 135 to 0. The total amount of the distortion, using the original modification rules table, which is shown in Table 5, is 1095. On the other hand, the total amount, using the new alter rules, is 715. Hence, the proposed scheme indeed effectively reduces image distortion.

The detailed process of the proposed scheme is described, as follows:Preprocessing:Set X to be the original image and s to be the secret message.Use two secret bits $m_{1}$ and $m_{2}$ of s and the corresponding pixel A of X to count the occurrence frequency.Compute LSB(A) and F(A, A) and $t=m_{1}\times 2+m_{2}$.Calculate the frequency by $\gamma_{F\left(A,\;A\right)}^{LSB\left(A\right)}\left(t\right)=\gamma_{F\left(A,\;A\right)}^{LSB\left(A\right)}\left(t\right)+1$.Map the embedding rule table, which is shown in Table 4, to find the corresponding modification rule.Calculate the distortion $\delta_{F\left(A,\;A\right)}^{LSB\left(A\right)}\left(t\right)={\left(A-A^{\prime }\right)}^{2}+{\left(A-A^{\prime \prime }\right)}^{2}$.Repeat (B)–(F) until all pixels have been processed.Rank the combinations by $\delta_{F\left(A,\;A\right)}^{LSB\left(A\right)}\left(t\right)$ in increasing order to obtain $\rho_{F\left(A,\;A\right)}^{LSB\left(A\right)}\left(t\right)$.Rank the combinations by $\gamma_{F\left(A,\;A\right)}^{LSB\left(A\right)}\left(t\right)$ in decreasing order to obtain ${\hat{\rho }}_{F\left(A,\;A\right)}^{LSB\left(A\right)}\left(t\right)$.Embedding Process:Rescan the image X and re-start from the first bit of s.Use two secret bits $m_{1}$ and $m_{2}$ of *s* and the corresponding pixel A of X to compute LSB(A), F(A, A) and $t=m_{1}\times 2+m_{2}$.Use the values LSB(A), F(A, A) and t to find the corresponding order ${\hat{\rho }}_{F\left(A,\;A\right)}^{LSB\left(A\right)}\left(t\right)$.Set $\varepsilon ={\hat{\rho }}_{F\left(A,\;A\right)}^{LSB\left(A\right)}\left(t\right)$.Find the rule $\left(L{A^{\prime }}_{F\left(A,\;A\right)}^{LSB\left(A\right)}\left(\varepsilon \right),L{A^{\prime \prime }}_{F\left(A,\;A\right)}^{LSB\left(A\right)}\left(\varepsilon \right)\;\right)$ from Table 6 to compute $A^{\prime }$ and $A^{\prime \prime }$.Repeat the steps (B)–(E) until all messages are embedded into the image.

Figure 6 shows an embedding example. The original image is X={44,45,37,…,5}. The first pixel is 44. Suppose that the secret data is $s={\left(10\right)}_{2}$. The scheme calculates the LSB function and *F* function to get LSB(44)=0 and $F\left(44,44\right)=LSB\left(\lfloor \frac{44}{2}\rfloor +44\right)=0$. Suppose that the final re-encoding table is shown in Table 6. The new code of the combination with LSB(A)=0, and F(A, A)=0 and $t=m_{1}\times 2+m_{2}=2$ is $\varepsilon ={\hat{\rho }}_{F\left(A,\;A\right)=0}^{LSB\left(A\right)=0}\left(2\right)=0$. The altered rules of ε=0 are $\left(L{A^{\prime }}_{F\left(A,\;A\right)=0}^{LSB\left(A\right)=0}\left(0\right),L{A^{\prime \prime }}_{F\left(A,\;A\right)=0}^{LSB\left(A\right)=0}\left(0\right)\;\right)=\left(+0,+0\right)$. Therefore, the stego-pixels are $A^{\prime }=44+0=44$ and $A^{\prime \prime }=44+0=44$.

The next pixel is 45, if the confidential message is $s={\left(11\right)}_{2}$. The function values are LSB(45)=1, and F(45,45)=1. The new code of the combination with LSB(A)=1, F(A,A)=1 and $t=m_{1}\times 2+m_{2}=3$ is ${\hat{\rho }}_{F\left(A,\;A\right)=1}^{LSB\left(A\right)=1}\left(3\right)=2$. The alter rule of $\varepsilon ={\hat{\rho }}_{F\left(A,\;A\right)=1}^{LSB\left(A\right)=1}\left(3\right)=2$ is $\left(L{A^{\prime }}_{F\left(A,\;A\right)=1}^{LSB\left(A\right)=1}\left(2\right),L{A^{\prime \prime }}_{F\left(A,\;A\right)=1}^{LSB\left(A\right)=1}\left(2\right)\;\right)=\left(+0,+1\right)$. The stego-pixels are $A^{\prime }=45+0=45$ and $A^{\prime \prime }=45+1=46$.

The two camouflage images along with the re-encoding table are sent to the receiver for further extracting and recovering.

### 3.3. Information Extraction and Image Restoration Phase

In this phase, the receiver starts the procedure after having received two camouflage images and the re-encoding table. This method for information extraction is like Tseng et al.’s scheme. The original pixel value *A* is restored by calculating the floor average of two camouflage pixels $A^{\prime }$ and $A^{\prime \prime }$.
(11)$$A=\lfloor \frac{A^{\prime }+A^{\prime \prime }}{2}\rfloor$$

The information extraction procedure can be implemented after recovering *A* because the scheme needs the original pixel value to do the calculation for some cases.

First, the scheme extracts the temp messages by using Equations (12) and (13).
(12)$${\hat{m}}_{1}=LSB\left(A^{\prime }\right)$$
(13)$${\hat{m}}_{2}=\{\begin{array}{cc} LSB\left(\lfloor \frac{A}{2}\rfloor +A^{\prime \prime }\right), & if\;\left|A^{\prime }-A^{\prime \prime }\right|=2,\\ LSB\left(\lfloor \frac{A^{\prime }}{2}\rfloor +A^{\prime \prime }\right), & otherwise. \end{array}$$

The messages obtained from Equations (12) and (13) are not the original messages. Hence, the scheme uses the symbols ${\hat{m}}_{1}$ and ${\hat{m}}_{2}$ to represent the temp messages. The formula for extracting ${\hat{m}}_{1}$ is obtained by calculating the LSB of the pixel value of the first camouflage image. There are two situations for extracting ${\hat{m}}_{2}$. If the distance between two camouflage pixels $A^{\prime }$ and $A^{\prime \prime }$ is equal to 2, then the embedding rule used in the pixel is $(A^{\prime }=A-1$ and $A^{\prime \prime }=A+1)$. The extracted message might be wrong when the extraction equation is equal to $LSB\left(\lfloor A^{\prime }/2\rfloor +A^{\prime \prime }\right)$. Hence, the equation is changed to $LSB\left(\lfloor A/2\rfloor +A^{\prime \prime }\right)$.

The order of the combination is recovered by the following equations.
(14)$$\rho ={\hat{m}}_{1}\times 2+{\hat{m}}_{2}$$

The scheme uses ρ to map the corresponding new order $\hat{\rho }$ from the re-encoding table. The corresponding messages are extracted by mapping $\hat{\rho }$ to $m_{1}$ and $m_{2}$. For example, suppose that A=44, $m_{1}=0$, and $m_{2}=1$. The function values are LSB(44)=0 and F(44,44)=0. The new code of the combination with LSB(44)=0 and F(44,44)=0 and $t=m_{1}\times 2+m_{2}=1$ is ${\hat{\rho }}_{F\left(A,\;A\right)=0}^{LSB\left(A\right)=0}\left(1\right)=3$. The alter rule of $\varepsilon ={\hat{\rho }}_{F\left(A,\;A\right)=0}^{LSB\left(A\right)=0}\left(1\right)=3$ is $\left(L{A^{\prime }}_{F\left(A,\;A\right)=0}^{LSB\left(A\right)=0}\left(3\right),L{A^{\prime \prime }}_{F\left(A,\;A\right)=0}^{LSB\left(A\right)=0}\left(3\right)\;\right)=\left(-1,+1\right)$. The stego-pixels are $A^{\prime }=44-1=43$ and $A^{\prime \prime }=44+1=45$.

In the extraction phase, the original pixel is recovered by $A=\lfloor \left(A^{\prime }+A^{\prime \prime }\right)/2\rfloor =\lfloor \left(43+45\right)/2\rfloor =44$. The first temp secret bit is computed by ${\hat{m}}_{1}=LSB\left(A^{\prime }\right)=LSB\left(43\right)=1$. Because $\left|A^{\prime }-A^{\prime \prime }\right|=\left|43-45\right|=2$, the second bit is calculated by ${\hat{m}}_{2}=LSB\left(\lfloor A/2\rfloor +A^{\prime \prime }\right)=LSB\left(\lfloor 44/2\rfloor +45\right)=1$. The order of the combination is $\rho ={\hat{m}}_{1}\times 2+{\hat{m}}_{2}=1\times 2+1=3$. The scheme uses LSB(A)=0 and F(A,A)=0 and ρ=3 to map the corresponding new order $\hat{\rho }$ from the re-encoding table, which is shown in Table 6. The corresponding messages are extracted by mapping $\hat{\rho }$ to its corresponding message $m_{1}=0$ and $m_{2}=1$. The diagram of the mapping is shown in Figure 7.

Similarly, let us take Figure 6 as an example. The first pixel is recovered by A=⌊(44+44)/2⌋=44. The temp messages are ${\hat{m}}_{1}=LSB\left(44\right)=0$ and ${\hat{m}}_{2}=LSB\left(\lfloor 44/2\rfloor +44\right)=0$. The order value is $\rho ={\hat{m}}_{1}\times 2+{\hat{m}}_{2}=0$. The corresponding new order $\hat{\rho }$ of the combination with LSB(44)=0 and F(A,A)=0 and ρ=0 is ${\hat{\rho }}_{F\left(A,\;A\right)=0}^{LSB\left(A\right)=0}\left(2\right)=0$, where t=2. The mapping messages are $m_{1}=1$ and $m_{2}=0$. The diagram of the mapping is shown in Figure 8.

The second pixel is $A=\lfloor \frac{45+46}{2}\rfloor =45$. The temp messages are ${\hat{m}}_{1}=LSB\left(45\right)=1$ and ${\hat{m}}_{2}=LSB\left(\lfloor 45/2\rfloor +46\right)=0$. The order value is ρ=1×2+0=2. The corresponding new order $\hat{\rho }$ of the combination with LSB(45)=1 and F(45,45)=1 and ρ=2 is ${\hat{\rho }}_{F\left(A,\;A\right)=1}^{LSB\left(A\right)=1}\left(t\right)=2$, where t=3. The mapping messages are $m_{1}=1$ and $m_{2}=1$. The final confidential message is $S={\left(1011\right)}_{2}$. The diagram of the embedding example is shown in Figure 9.

### 3.4. Overflow and Underflow Problem

The method proposed in this study is an improvement on the method suggested by Tseng et al.’s scheme. Therefore, the same overflow and underflow problems will exist. To solve this problem and avoid these issues, we could apply the same rules as the method proposed by Tseng et al. If the original pixel value is equal to 0 or 255, then the pixel is non-embeddable and cannot be used to conceal the message. For the non-embeddable pixel, the value will remain unchanged. In the extraction process, if both camouflage pixels are equal to 0 or 255, then it means there is no secret data hidden in it. Even if only one stego-pixel value is equal to 0 or 255 it is evident that it still has the secret message concealed in it.

## 4. Experimental Results and Discussions

This research employed Matlab (version R2016b) as the test environment, and used six grayscale images (512 × 512 in size) to conduct experiments in order to verify whether the research method can effectively improve the image quality. The following figures show the six grayscale images used in the experiment. The test images are Lena, Mandrill, Pepper, Airplane, Lake, and Tiffany. The images are shown in Figure 10.

In the experiment, the Peak Signal-to-Noise Ratio (PSNR) is used to quantify the image quality. PSNR is usually expressed as a logarithmic quantity using the decibel (dB) scale. The following is the definition of PSNR:(15)$$PSNR=10\times \left(\frac{255^{2}}{MSE}\right)\;\left(dB\right)$$

The MSE of the equation is the mean square error, which represents the square root of the difference between the original image and the camouflage image. Therefore, the smaller the MSE value, the better the calculated image quality. The MSE formula is as follows:(16)$$MSE=\frac{1}{h\times w}\overset{h}{\underset{i=1}{\sum }}\overset{w}{\underset{j=1}{\sum }}{\left(x_{i,j}-x_{i,j}^{\prime }\right)}^{2}$$
*h* and *w*, respectively, represent the length and width of the image, $x_{i,j}$ represents the pixels of the original image, and $x_{i,j}^{\prime }$ represents the pixels of the camouflage image. Dual-image technology is implemented in this paper. Hence, we used the average of the two camouflage-image PSNR values in our comparison with other methods.

The hiding payload is computed by:(17)$$\mathrm{bpp}=\frac{TB}{2\times h\times w}.$$
where *TB* is the total number of secret message bits, which are embedded into the cover image (and “bpp” means bits per pixel). Because the proposed scheme is a dual-image based scheme, the total number of pixels is 2 × *h* × *w*.

This study compares the proposed scheme with Lee et al.’s scheme, schemes Lee2009 and Lee2013 [6,7], Qin and Chang et al.’s scheme [3], the center folding scheme (CenterFolding) proposed by Lu et al. [9], LSB-M proposed by Lu et al. [24], and LSB-MA proposed by Tseng et al. [25] The proposed scheme is indicated with LSB-MA-ordering. Figure 11, Figure 12 and Figure 13 show the experimental results.

In the first experiment, the scheme randomly generates the secret message and conceals it into the test image. Figure 11 shows the results of concealing the random bits into Lena. The hiding payloads of Chang2013 and Center Folding are higher than that of the others, however, the PSNR values of Chang2013 and CenterFolding are worse than that of the others. The LSB matching based schemes can achieve higher image quality. Among them, the hiding payload of Lee2009 is the least. The image quality of LSB-MA-ordering is the highest.

From Figure 11 and Figure 12 we can observe that the proposed scheme can get the highest image quality regardless of whether the image is complex or smooth. The experimental results of the other images have very similar curve to Lena. Hence, this study only shows the results of two images: Lena and Mandrill.

In the following experiment, this study used an image as the secret message to be concealed into the test image. Figure 13 shows the experimental results. The image quality of the proposed scheme is still the highest among the comparison schemes.

To test the efficient of the proposed scheme, this study statistics the probability of different pixels with different secret message by using three different types of secret messages. The first secret message was generated by the random number generator (RNG), the second and the third message were the fixed images ‘Tiffany’ and ‘CYUT Bird’, which are shown in Figure 14. The statistic results are shown in Table 7. There are 16 different combinations composed by LSB(A), F(A, A), $m_{1}$ and $m_{2}$. The column ‘Count’ is the total number of the occurrence of the combination, ‘%’ is the percentage of the combination, $LA^{\prime }$ and $LA^{\prime \prime }$ are the operator to compute the stego-pixels $A^{\prime }$ and $A^{\prime \prime }$, ‘δ’ is the distortion made by $LA^{\prime }$ and $LA^{\prime \prime }$, and ‘TD’ is the total number of distortion made by the combination. ‘Total Distortion’ in the end of the table is the total distance made by the secret image.

Form the table we can see that the count of each combination of the random number generate is almost the same. The average percentage is 6.25% and the standard deviation is 0.00049. The total number of the distortion made by the random number generate is 263,040. The count of the secret image ‘Tiffany’ is different from that of the RNG. The average percentage to embed $m_{1}=1$ and $m_{2}=1$ is 9.01%. The standard deviation is 0.02209. The total distortion of the secret image ‘Tiffany’ is 208,143, which is smaller than that of the RNG.

The count of each combination of the secret image ‘CYUT Bird’ is very different from the other two secret messages. Because the secret image has a lot of white pixels, which most significant bits start with ‘11’ in the binary system. Hence, the probability of the pixel concealed with $m_{1}=1$ and $m_{2}=1$ is very high. The average percentage to embed $m_{1}=1$ and $m_{2}=1$ is 15.95%. The standard deviation is 0.05608. The total distortion of the secret image is 121,120, which is smallest of the three secret messages. Therefore, the image quality can be greatly improved especially for the secret image, which is not uniform.

Furthermore, although the proposed scheme needs to transmit the mapping table to the receiver for extracting the secret message and recovering the original image, the necessary extra information that needed to be transmitted is $\hat{\rho }$ and ρ of the mapping table. The scheme transforms the numbers of $\hat{\rho }$ into the binary system to form a binary string. There are 16 numbers of $\hat{\rho }$. The maximum value of $\hat{\rho }$ is 3. Hence, the scheme only uses two bits to encode the value. After encoded the total length of the binary string of $\hat{\rho }$ is 16×2=32 bits.

The values of ρ are also encoded to a binary string. The length of the string is also 16×2=32. Two binary strings are concatenated together to transmit to the receiver. 

The receiver picks two bits up from the binary strings each time and transforms the bits into a decimal number to fill into the mapping table. If the value of $\hat{\rho }$ is equal to 0, then $LA^{\prime }=0$ and $LA^{\prime \prime }=0$. If the value of $\hat{\rho }$ is equal to 1, then $LA^{\prime }=0$ and $LA^{\prime \prime }=1$. If the value of $\hat{\rho }$ is equal to 2, then $LA^{\prime }=1$ and $LA^{\prime \prime }=0$. If the value of $\hat{\rho }$ is equal to 3 then $LA^{\prime }=-1$ and $LA^{\prime \prime }=1$.

Therefore, the total number of the mapping table sent from the sender is 32+32=64 (bits).

In 2001, Fridrich et al. [27] proposed the use of RS steganalysis to detect whether an image has a secret message in it [28]. The technology uses a judgment function and a flipping function to divide the pixels into two groups. The judgment function separates the groups into two types, based on smoothness and regularity. The flipping function segregates the groups into three categories: Regular (R), Singular (S), and Unusable (U). The technology applies two masks M = [1 0 0 1] and -M = [−1 0 0 −1] to calculate the percentages of regular, singular, and unusable that are marked by R_M_G, R_FM_G, S_M_G, S_FM_G, U_M_G, and U_FM_G, respectively. The hypotheses are R_M_G ≅ R_FM_G, S_M_G ≅ S_FM_G, and U_M_G ≅ U_FM_G. Figure 15 and Figure 16 are the RS steganalysis results of Mandrill.

From the figures, we can see that the curve of R_M_G is close to that of R_FM_G. The curves of S_M_G, S_FM_G, U_M_G, and U_FM_G have the same shapes. Therefore, the proposed scheme can against the RS steganalysis.

## 5. Conclusions and Future Works

In this paper, a modified LSB matching method using the dual-image and likelihood recording strategy is proposed. The scheme analyzes all possible modifications under all hidden conditions and re-encodes each combination according to the frequency of its occurrence. The combination with a higher occurrence rate is re-encoded with a lower modification rule. The experimental results show that the embedding capacity of the proposed method is like the method used in Tseng et al.’s scheme. Moreover, the image quality of the proposed scheme is the highest among the comparison methods. Furthermore, the proposed scheme against the steganalysis attacks.

The reason why the proposed scheme can get higher image quality is because the scheme transforms the worst cases with better encode results. In general, each worst case may make two images distance where one pixel is -1 and the other one is +1. That means the case will generate $2^{2}=4$ squared errors for computing the image quality. In the proposed scheme, the worst case is re-encoded with the minimum distortion code 0. Hence, the stego-pixel is the same with the original one such that the image distance is 0. The image distortion can be effectively reduced. The proposed scheme is especially suitable for simple secret image such as logo, cartoons, and signature. On the contrary, the proposed scheme can only get a few increasing while the secret image is a uniform image.

In the future, we will try to figure out how to re-encode the secret image according to its characteristic. The scheme needs adaptively or elastically change the encoding strategy such as running encoding or Huffman coding, to encode the different cases to further reduce the image distortion, or filter the bad cases, which will cause huge damage in the pre-processing procedure. Furthermore, add more translation tables to improve the image quality.

## Figures and Tables

**Figure 1 entropy-23-00577-f001:**
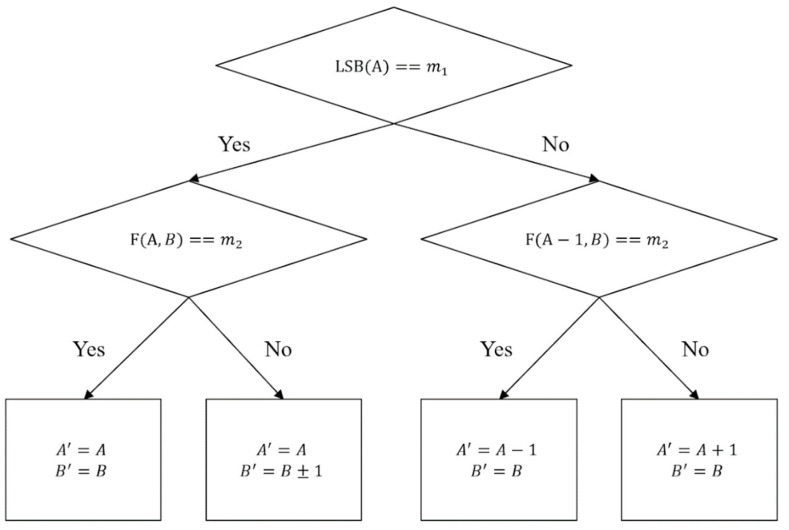
LSB matching hiding flowchart.

**Figure 2 entropy-23-00577-f002:**
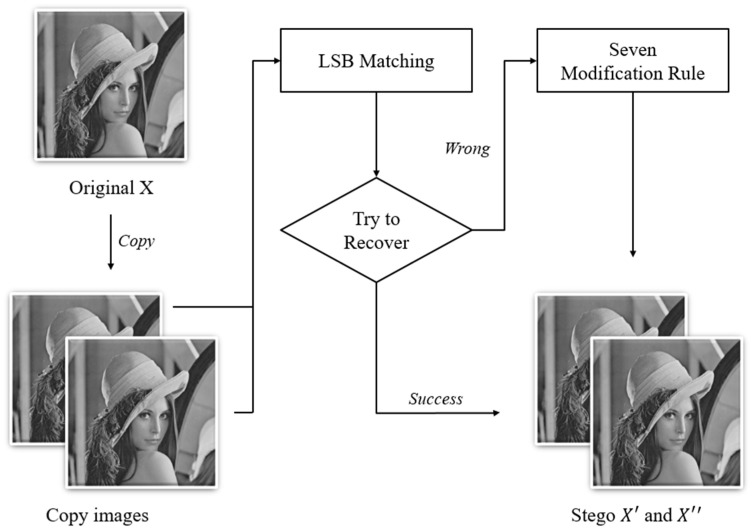
The flowchart of the dual-image LSB matching method proposed by Lu et al.

**Figure 3 entropy-23-00577-f003:**
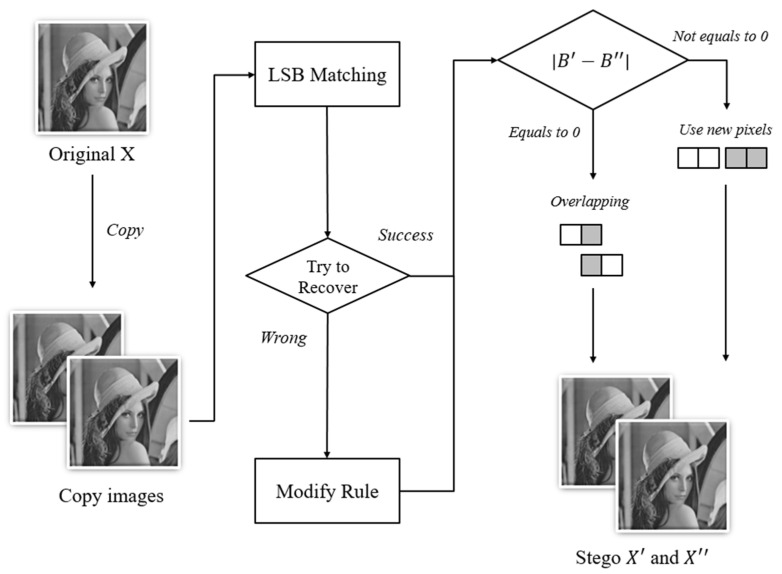
The flowchart of the dual-image LSB matching method by Wang et al.

**Figure 4 entropy-23-00577-f004:**
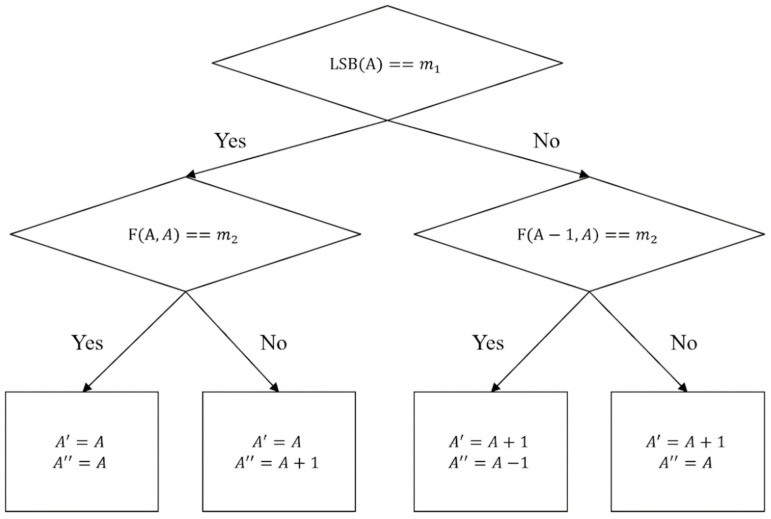
Dual-image modified LSB matching hiding flowchart by Tseng et al.

**Figure 5 entropy-23-00577-f005:**
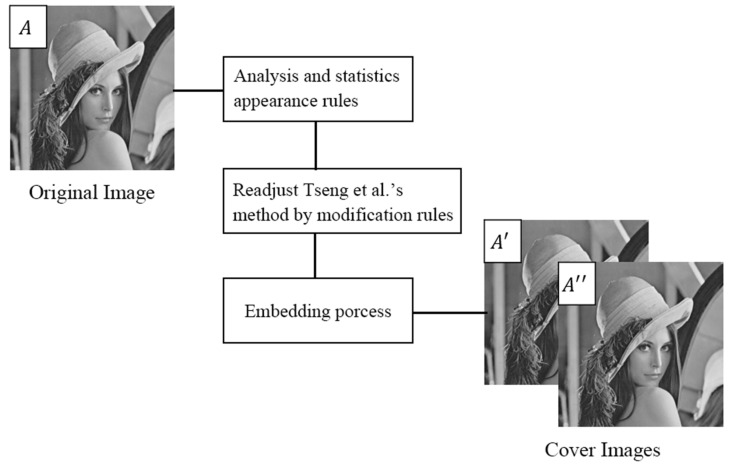
Diagram of the proposed method.

**Figure 6 entropy-23-00577-f006:**
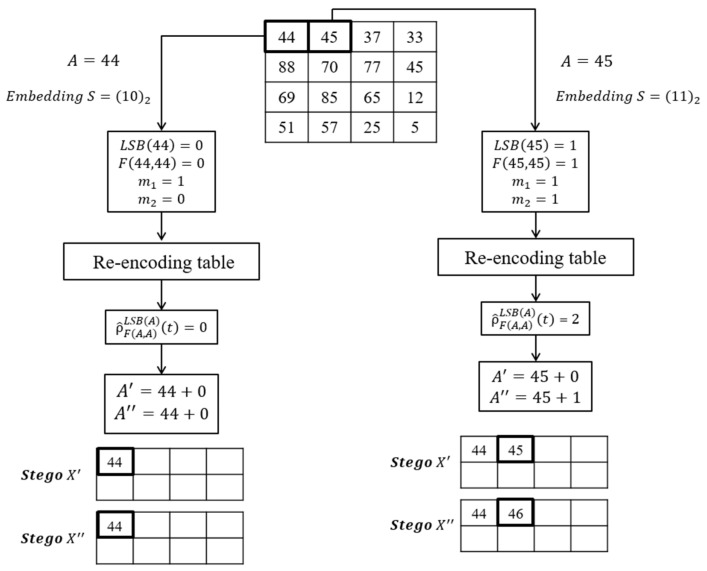
An embedding example.

**Figure 7 entropy-23-00577-f007:**
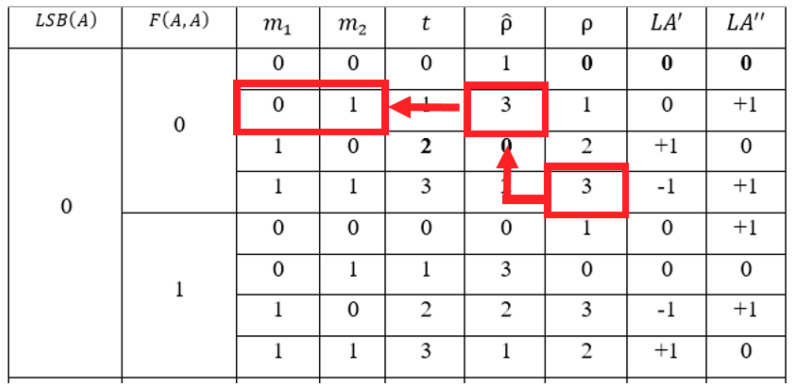
The mapping of ρ=3 to its corresponding $\hat{\rho }$ and messages $m_{1}$ and $m_{2}$.

**Figure 8 entropy-23-00577-f008:**
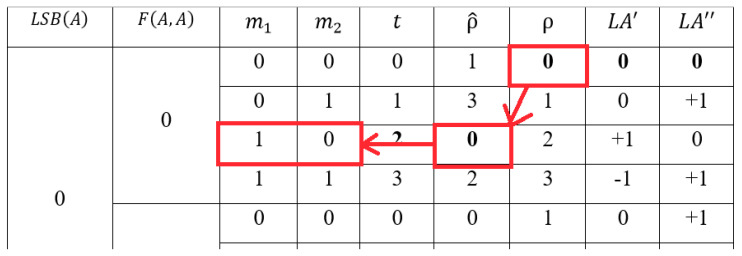
The mapping of ρ=0 to its messages $m_{1}=1$ and $m_{2}=0$.

**Figure 9 entropy-23-00577-f009:**
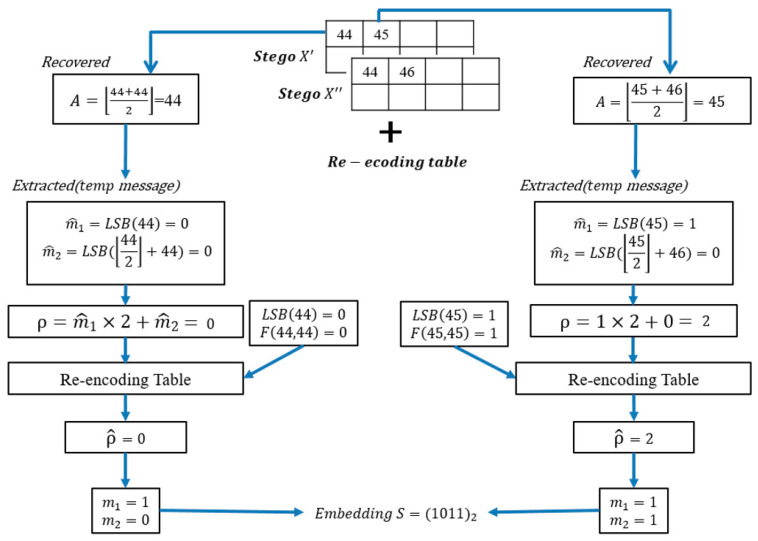
The extraction example of Figure 6.

**Figure 10 entropy-23-00577-f010:**
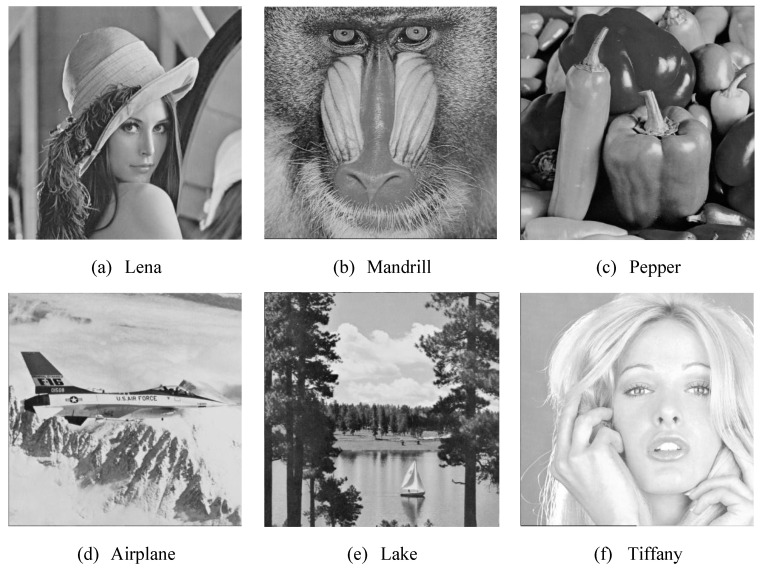
The six test images (**a**) Lena (**b**) Mandrill (**c**) Pepper (**d**) Airplane (**e**) Lake (**f**) Tiffany each of them has different characteristics.

**Figure 11 entropy-23-00577-f011:**
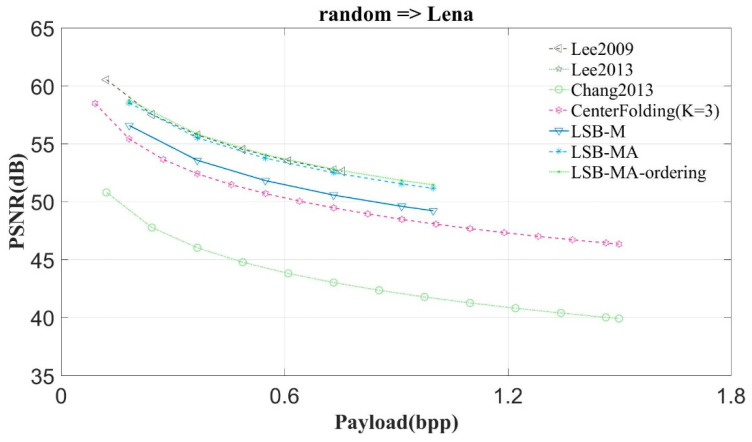
The experimental results of Lena.

**Figure 12 entropy-23-00577-f012:**
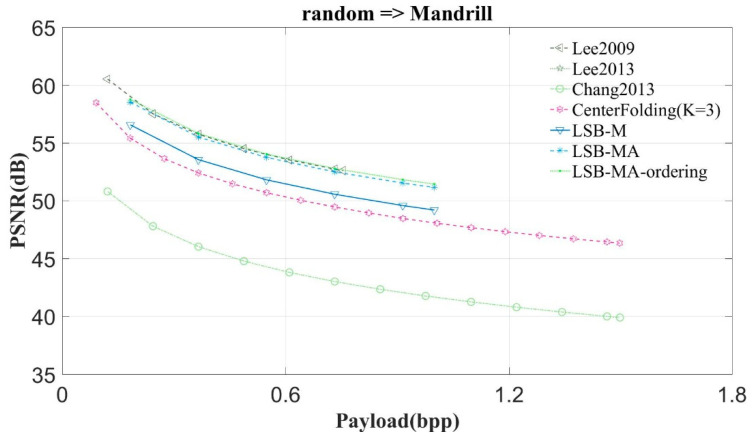
The experimental results of Mandrill.

**Figure 13 entropy-23-00577-f013:**
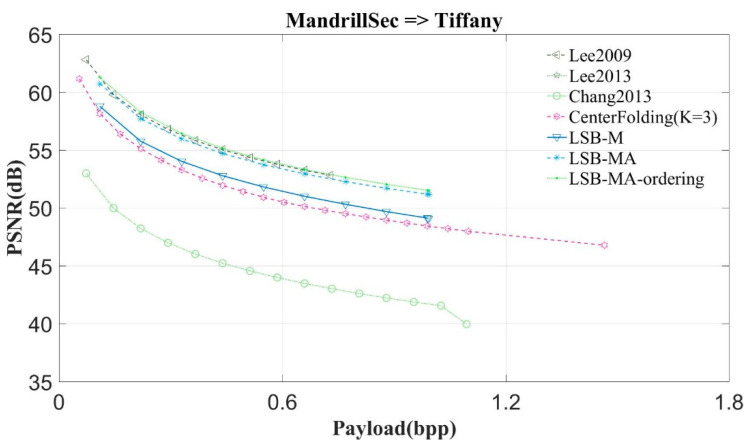
The experimental results of Tiffany with the secret image Mandrill.

**Figure 14 entropy-23-00577-f014:**
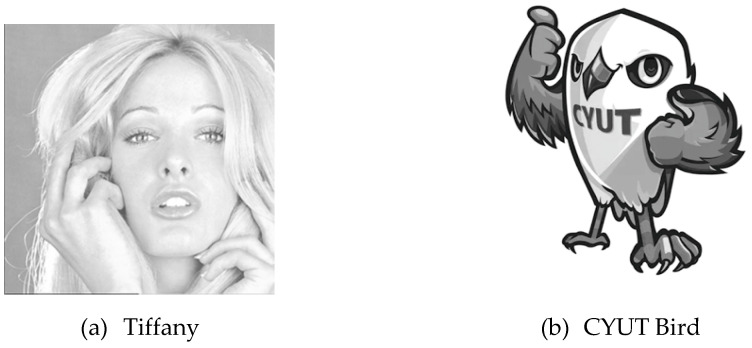
The fixed secret images ‘Tiffany’ and ‘CYUT Bird’.

**Figure 15 entropy-23-00577-f015:**
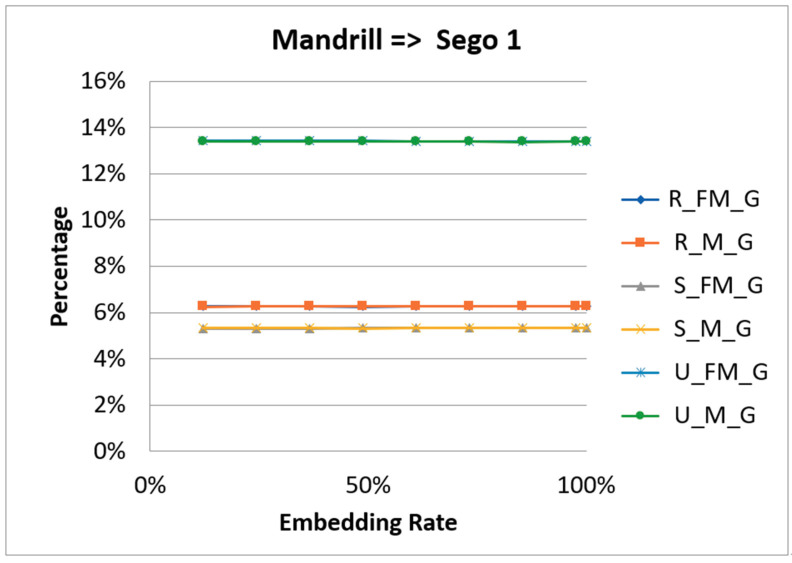
The RS analysis result of the first stego image of Mandrill.

**Figure 16 entropy-23-00577-f016:**
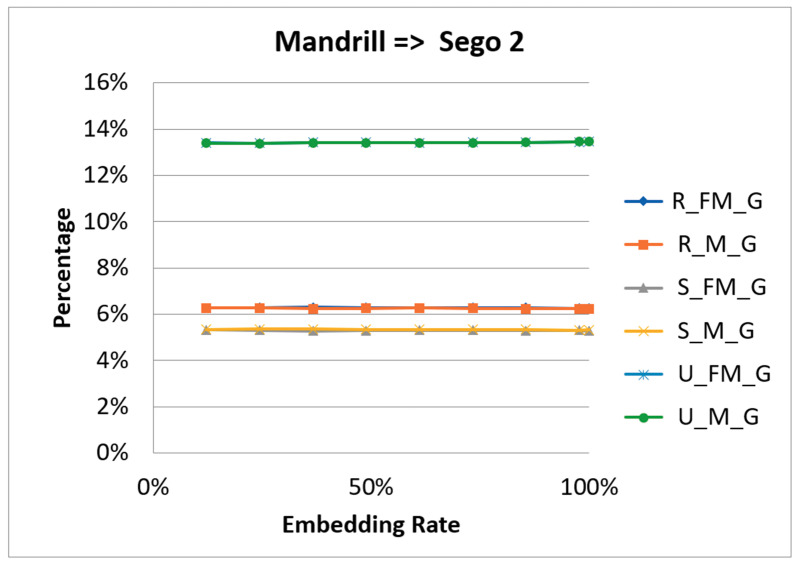
The RS analysis result of the second stego image of Mandrill.

**Table 1 entropy-23-00577-t001:** The seven Modification Rules designed by Lu et al.

Rule	Pixel Value Condition				Final Adjustment			
	$TA^{\prime }$	$TB^{\prime }$	$TA^{\prime \prime }$	$TB^{\prime \prime }$	$A^{\prime }$	$B^{\prime }$	$A^{\prime \prime }$	$B^{\prime \prime }$
1	0	0	−1	0	A+2	B+1	A−1	B+1
2	0	1	0	1	A	B+1	A	B−1
3	0	1	−1	0	A+2	B	A−1	B
4	−1	0	0	0	A−1	B	A+2	B+1
5	−1	0	0	1	A−1	B	A+2	B
6	−1	0	−1	0	A−1	B+2	A+1	B−1
7	1	0	1	0	A−1	B−1	A+1	B+2

**Table 2 entropy-23-00577-t002:** The Modify Rules Table designed by Wang et al.

Case	Pixel Value Condition				Final Adjustment			
	$TA^{\prime }$	$TB^{\prime }$	$TA^{\prime \prime }$	$TB^{\prime \prime }$	$A^{\prime }$	$B^{\prime }$	$A^{\prime \prime }$	$B^{\prime \prime }$
1	0	0	−1	0	A+2	B+3.	A−1	B−2
2	0	1	0	1	A	B+3	A	B−3
3	0	1	−1	0	A+2	B−1	A−2	B+2
4	−1	0	0	0	A−1	B−2	A+2	B+3
5	−1	0	0	1	A−1	B+2	A+2	B−2
6	−1	0	−1	0	A−1	B+4	A+1	B−3
7	1	0	1	0	A−1	B−3	A+1	B+4
8	0	0	0	1	A	B−2	A	B+3
9	1	0	0	1	A+1	B−2	A	B+3
10	0	1	1	0	A	B+3	A+1	B−2
11	0	1	0	0	A	B+3	A	B−2

**Table 3 entropy-23-00577-t003:** Embedding Rules Table of the proposed scheme.

A	LSB(A)	F(A, A)	$m_{1}$	$m_{2}$	$A^{\prime }$	$A^{\prime \prime }$	Distortion(δ)	Ordering(ρ)
Even	0	0	0	0	0	0	0	0
			0	1	0	+1	1	1
			1	0	+1	0	1	2
			1	1	−1	+1	2	3
		1	0	0	0	+1	1	1
			0	1	0	0	0	0
			1	0	−1	+1	2	3
			1	1	+1	0	1	2
Odd	1	1	0	0	+1	0	1	1
			0	1	−1	+1	2	3
			1	0	0	+1	1	2
			1	1	0	0	0	0
		0	0	0	−1	+1	2	3
			0	1	+1	0	1	1
			1	0	0	0	0	0
			1	1	0	+1	1	2

**Table 4 entropy-23-00577-t004:** The Re-encoding Table of the proposed scheme.

A	LSB(A)	F(A, A)	$m_{1}$	$m_{2}$	t	γ	δ	ρ	$\hat{\rho }$
Even	0	0	0	0	0	$\gamma_{0}^{0}\left(0\right)$	$\delta_{0}^{0}\left(0\right)$	$\rho_{0}^{0}\left(0\right)$	${\hat{\rho }}_{0}^{0}\left(0\right)$
			0	1	1	$\gamma_{0}^{0}\left(1\right)$	$\delta_{0}^{0}\left(1\right)$	$\rho_{0}^{0}\left(1\right)$	${\hat{\rho }}_{0}^{0}\left(1\right)$
			1	0	2	$\gamma_{0}^{0}\left(2\right)$	$\delta_{0}^{0}\left(2\right)$	$\rho_{0}^{0}\left(2\right)$	${\hat{\rho }}_{0}^{0}\left(2\right)$
			1	1	3	$\gamma_{0}^{0}\left(3\right)$	$\delta_{0}^{0}\left(3\right)$	$\rho_{0}^{0}\left(3\right)$	${\hat{\rho }}_{0}^{0}\left(3\right)$
		1	0	0	0	$\gamma_{1}^{0}\left(0\right)$	$\delta_{1}^{0}\left(0\right)$	$\rho_{1}^{0}\left(0\right)$	${\hat{\rho }}_{1}^{0}\left(0\right)$
			0	1	1	$\gamma_{1}^{0}\left(1\right)$	$\delta_{1}^{0}\left(1\right)$	$\rho_{1}^{0}\left(1\right)$	${\hat{\rho }}_{1}^{0}\left(1\right)$
			1	0	2	$\gamma_{1}^{0}\left(2\right)$	$\delta_{1}^{0}\left(2\right)$	$\rho_{1}^{0}\left(2\right)$	${\hat{\rho }}_{1}^{0}\left(2\right)$
			1	1	3	$\gamma_{1}^{0}\left(3\right)$	$\delta_{1}^{0}\left(3\right)$	$\rho_{1}^{0}\left(3\right)$	${\hat{\rho }}_{1}^{0}\left(3\right)$
⋮	⋮	⋮	⋮	⋮		⋮	⋮	⋮	⋮
…	…	…	…	…		$\gamma_{F\left(A,\;A\right)}^{LSB\left(A\right)}\left(t\right)$	$\delta_{F\left(A,\;A\right)}^{LSB\left(A\right)}\left(t\right)$	$\rho_{F\left(A,\;A\right)}^{LSB\left(A\right)}\left(t\right)$	${\hat{\rho }}_{F\left(A,\;A\right)}^{LSB\left(A\right)}\left(t\right)$

**Table 5 entropy-23-00577-t005:** A Re-encoding Table example.

A	LSB(A)	F(A, A)	t	$m_{1}$	$m_{2}$	γ	δ	ρ	θ	$\hat{\delta }$	$\hat{\rho }$	$\hat{\theta }$
Even	0	0	0	0	0	50	0	0	0	1	1	50
			1	0	1	40	1	1	40	2	3	80
			2	1	0	135	1	2	135	0	0	0
			3	1	1	45	2	3	90	1	2	45
		1	0	0	0	120	1	1	120	0	0	0
			1	0	1	50	0	0	0	2	3	100
			2	1	0	60	2	3	120	1	2	60
			3	1	1	80	1	2	80	1	1	80
Odd	1	1	0	0	0	110	1	1	110	1	1	110
			1	0	1	130	2	3	260	0	0	0
			2	1	0	10	1	2	10	2	3	20
			3	1	1	50	0	0	0	1	2	50
		0	0	0	0	30	2	3	60	1	2	30
			1	0	1	20	1	1	20	2	3	40
			2	1	0	100	0	0	0	0	0	0
			3	1	1	50	1	2	50	1	1	50
Total distortion									1095			715

**Table 6 entropy-23-00577-t006:** The final Re-encoding Table.

LSB(A)	F(A, A)	$m_{1}$	$m_{2}$	t	$\hat{\rho }$	ρ	$LA^{\prime }$	$LA^{\prime \prime }$
0	0	0	0	0	1	0	0	0
		0	1	1	3	1	0	+1
		1	0	2	0	2	+1	0
		1	1	3	2	3	−1	+1
	1	0	0	0	0	1	0	+1
		0	1	1	3	0	0	0
		1	0	2	2	3	−1	+1
		1	1	3	1	2	+1	0
1	1	0	0	0	1	1	+1	0
		0	1	1	0	3	−1	+1
		1	0	2	3	2	0	+1
		1	1	3	2	0	0	0
	0	0	0	0	2	3	−1	+1
		0	1	1	3	1	+1	0
		1	0	2	0	0	0	0
		1	1	3	1	2	0	+1

**Table 7 entropy-23-00577-t007:** The statistic results of the proposed used the random number and the fixed secret images ‘CYUT Bird’ and ‘Tiffany’.

LSB(A)	F(A, A)	$m_{1}$	$m_{2}$	Random Number Generator (RNG)						Secret Image: Tiffany						Secret Image: CYUT Bird					
				Count	%	$LA^{\prime }$	$LA^{\prime \prime }$	δ	TD	Count	%	$LA^{\prime }$	$LA^{\prime \prime }$	δ	TD	Count	%	$LA^{\prime }$	$LA^{\prime \prime }$	δ	TD
0	0	0	0	16,180	6.17%	0	0	0	-	10,069	3.84%	−1	1	2	20,138	9191	3.51%	0	1	1	9191
		0	1	16,362	6.24%	0	1	1	16,362	11,265	4.30%	1	0	1	11,265	8040	3.07%	1	0	1	8040
		1	0	16,369	6.24%	1	0	1	16,369	20,508	7.82%	0	1	1	20,508	6544	2.50%	−1	1	2	13,088
		1	1	16,534	6.31%	−1	1	2	33,068	23,603	9.00%	0	0	0	-	41,670	15.90%	0	0	0	-
	1	0	0	16,235	6.19%	0	1	1	16,235	10,110	3.86%	−1	1	2	20,220	9137	3.49%	0	1	1	9137
		0	1	16,263	6.20%	0	0	0	-	11,445	4.37%	1	0	1	11,445	7934	3.03%	1	0	1	7934
		1	0	16,431	6.27%	−1	1	2	32,862	20,468	7.81%	0	1	1	20,468	6575	2.51%	−1	1	2	13,150
		1	1	16,463	6.28%	1	0	1	16,463	23,369	8.91%	0	0	0	-	41,746	15.92%	0	0	0	-
1	1	0	0	16,506	6.30%	1	0	1	16,506	10,237	3.91%	−1	1	2	20,474	9148	3.49%	0	1	1	9148
		0	1	16,585	6.33%	−1	1	2	33,170	11,283	4.30%	1	0	1	11,283	8298	3.17%	1	0	1	8298
		1	0	16,444	6.27%	0	1	1	16,444	20,434	7.79%	0	1	1	20,434	6560	2.50%	−1	1	2	13,120
		1	1	16,293	6.22%	0	0	0	-	23,874	9.11%	0	0	0	-	41,822	15.95%	0	0	0	-
	0	0	0	16,355	6.24%	−1	1	2	32,710	10,112	3.86%	−1	1	2	20,224	9150	3.49%	0	1	1	9150
		0	1	16,257	6.20%	1	0	1	16,257	11,218	4.28%	1	0	1	11,218	7836	2.99%	1	0	1	7836
		1	0	16,273	6.21%	0	0	0	-	20,466	7.81%	0	1	1	20,466	6514	2.48%	−1	1	2	13,028
		1	1	16,594	6.33%	0	1	1	16,594	23,683	9.03%	0	0	0	-	41,979	16.01%	0	0	0	-
Total distortion:				263,040						208,143						121,120					

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
