# Peer review of "Improving the Reversible LSB Matching Scheme Based on the Likelihood Re-Encoding Strategy"

_entropy, 2021, doi:10.3390/e23050577_

Round 1
Reviewer 1 Report
Lu et al. presented their work on dual image reversible embedding method based on modified least significant bit matching. this is modified method on previous approach that analyses every possible modifications in hidden circumstances and re-encodes back according to occurrence. the produced image quality is better compared to other similar methods. the work is explained clearly and in general free from any potential errors. this could be recommended for publication.
Author Response
We are thankful for the reviewer’s comments.

Reviewer 2 Report
The paper deals with an interesting problem: when embedding secret data using dual image steganography, can the distortion be decreased by a smart choice of the encoding. The paper offers such a tweaking and evaluates its efficiency.
There are many interesting problems in this area, and the paper shows a promising first step towards recognizing and attacking those problems. Nevertheless I see several aspects which require significant improvement. First, the suggested method bases not on entropy rather than probability frequencies. The error function to be minimized is the square distance rather than some entropy-like among such that the Kullback-Leibler difference. While I do not object the title, the description around line 331 is misleading.
The quite lengthy Section 2 reviews earlier embedding methods, which are also to be compared to the proposed modification. Unfortunately, some of the descriptions are cryptic, and seem to miss the main point of such a scheme. For example, Chen et al method (first scheme 2.1) as modified by Mielikainen, modifies only one of two neighboring pixels while still embedding two secret bits. Similarly, some of the the dual images methods are designed on similar ground. Maybe a more overall description rather than how bits are manipulated would help the reader to see what the different methods do. In particular, I was unable to decipher how methods described in 2.5, 2.6 and 2.7 work at all. See also some of the detailed remarks below.
The system considered in the paper is a dual image steganography where the same image is modified twice, and the embedded secret message can be recovered using both images. In addition, the original image can be recovered as the truncated average of the two stego images. The original method studied changes both images by at most 1 at each pixel and encodes two secret bits at each pixel position, thus this method cannot be improved on in this respect. Decoding works as follows: for each pixel position depending on the pixel values in the two images four cases might occur: the values are equal, the first one is one more than the second one, the second one is one more that the first one, and finally, the first one is two less than the second one. These cases correspond, in some order, to the secret bit pairs 00, 01, 10, and 11. This correspondence is determined by the last two bits of the original pixel value (which is recoverable from the stego images without extracting the secret information first), thus there are four of such mappings. The proposal tries to minimize the number of cases when the modified pixel values differ by 2. It is done by redefining the four maps above so that the worst case is used a minimal number of times. The new maps should be transferred to the decoder additionally to the stego images.
Such a description of the method would definitely help the reader to understand better what is going on, and also pinpoint several questions and possible improvements.
1) It is clear that if the secret is a random bit sequence, then some, but not too much improvement can be expected. A 512 by 512 image can embed 2^18 bit pairs, one fourth in each translation table. Thus a single table is applied to approximately 2^16 plus minus 2^8 pairs, (the expected deviation is 2^8=256), thus we expect improvement (creating difference one instead of difference two) about 4*256 times, which is around 1000 pixels. On the other hand, if the embedded image is not uniform (e.g. they are the most significant upper two bits of another image) then the improvement could be much bigger. It would be interesting to see statistics on these numbers.
2) The modification requires to transmit the modified map on the clear. Instead, it could be interesting to use a "running encoding". Starting from the standard maps, after encountering some secret bits, the maps are modified according to the observed frequency so far. Even this running length can be fixed (or rather than running length, it can be taken as an area of the processed part of the image). This corresponds to the running Huffman coding used by compression softwares. Even potentially it can even reduce the distortion by more than the method suggested by the authors.
3) Allowing reduction in the secret data, all bad cases can be discarded, improving the situation to the best possible. If it is not so desirable, there are fancy coding techniques which translate the original secret stream into another stream of bit pairs having relatively small number of "bad" cases (on the average). Such coding can be applied as a pre- and postprocessing without requiring additional data transfer next to the images, and also improve image quality.
4) Allowing more translation tables (depending on more bits of the actual pixel, or on bits of neighboring pixels) could also improve the quality.
Some detailed remarks.
Lines 183-195: The description should use infinite form as it describes what to do, and not what the algorithm actually performs, such as "Duplicates" => "duplicate", "Uses" => "use", etc.
Line 191: "On the contrary" => "otherwise" (they mean different things)
Line 203: "if equation (5) equals 0 or lower than 3" does not make sense compared to what equation (5) is. Should it be zero, or should it be less than 3?
Line 204: "Ostensibly" does not mean what was intended, perhaps it should be "Otherwise"?
Line 209: "which are not able" => "which are not recoverable"
Line 210: "be a larger distortion than that produced" => "larger than the one produced"
Line 211: "by reforming" => "by changing" or "by redesigning"
Line 230: I could not decipher this method.
Line 264: Neither these methods.
Line 270: There is no difference between formulas (16) and (17).
Lines 296-299: no need to move from binary to decimal and then to base 5. Go directly to base 5 systems.
Line 305: It is not clear what this method does.
Line 331: As indicated above, the method uses frequencies, not entropy.
Line 367: Same remark as above.
Line 382: In Table 5 the values in the brackets are form 0 to 3, not from 1 to 4.
Lines up to 512: Maybe a shorter description of the method (ideas, not bit crunching) would help the reader to understand what is going on.
Lines 558: My guess is that two such an examples suffice, as all are very similar.
Lines 580: Figure 23 and Figure 24 are superfluous. They do not show anything, and do not help. The difference is too small. If you insist on images, then plot the difference.
Line 601: This sentence has no verb.
